# Human Colonic Microbiota and Short-Term Postoperative Outcomes in Colorectal Cancer Patients: A Pilot Study

**DOI:** 10.3390/microorganisms10010041

**Published:** 2021-12-26

**Authors:** Lelde Lauka, Iradj Sobhani, Francesco Brunetti, Denis Mestivier, Nicola de’Angelis

**Affiliations:** 1Department of Digestive and Hepatobiliary Surgery, APHP-Henri Mondor University Hospital, 94000 Creteil, France; francesco.brunetti@aphp.fr (F.B.); nic.deangelis@yahoo.it (N.d.); 2EC2M-EA7375 Research Team, Henri Modor Campus, Paris East University, 94000 Creteil, France; denis.mestivier@u-pec.fr; 3Department of Gastroenterology, APHP-Henri Mondor University Hospital, 94000 Creteil, France

**Keywords:** microbiota, colon, colorectal cancer, surgery, postoperative complications

## Abstract

Despite the advances in surgical techniques and perioperative care, the complication rates after colorectal cancer surgery have remained stable. Recently, it has been suggested that colon microbiota may be implicated in several pathways that can lead to impaired colonic homeostasis and, thereby, to the development of complications after colorectal surgery. The aim of this study was to evaluate the potential impact of colonic dysbiosis on postoperative course. This prospective human clinical study recruited patients operated on for left colon, sigmoid colon or rectal cancer. Colon mucosa and fecal samples were collected to study mucosa associated microbiota (MAM) and luminal microbiota (LM), accordingly. Preliminary analysis for the first 25 consecutive patients with V3–V4 16S rRNA metagenomic analysis was performed. Bacterial composition and abundance in patients who developed postoperative complications over a 90-day follow-up period were compared to those without postoperative complications. Abundance and distribution of genera in MAM differed significantly when compared to LM with a significant impact on neoadjuvant therapy on bacterial composition. Preliminary analysis revealed no statistically significant differences in LM nor in MAM composition when individuals with and without postoperative surgical complications were compared. In cases of postoperative complications, LM and MAM showed significantly decreased diversity. Composition of the colonic microbiota is altered by neoadjuvant therapy. Results on the impact of colonic dysbiosis on postoperative complications are pending the end of the present study, with 50 patients enrolled.

## 1. Introduction

Colorectal cancer (CRC) is the third most common cancer worldwide with the second highest mortality in both sexes combined. Recently, a worrying trend of increasing early-onset CRC has been observed [1]. Postoperative complications in colorectal surgery are associated with increased mortality, increased readmission rate, longer hospital stay and, in cases of colorectal cancer, decreased overall and cancer-specific survival [2,3]. Despite the advances in surgical techniques and improvements in perioperative care over the last decade, the complication rates did not significantly decrease, particularly for anastomotic leakage occurrence [4].

In addition to the well-known risk factors, such as obesity, male gender, preoperative malnutrition, neoadjuvant therapy and low rectal anastomosis [5], recent studies focused on intestinal microbiota changes (dysbiosis) as a possible contributor to the development of postoperative complications with speculative mechanisms, as suggested by experimental studies in animals relating to infection [6], anastomotic leakage [3,7,8] and postoperative ileus [7]. Microbiota studies with human samples have suggested that it may be implicated in several pathways leading to impaired colonic homeostasis after colorectal surgery [8] and, potentially, in the development of complications, such as anastomotic leakage and postoperative ileus [9,10,11,12]. However, these few studies have each analyzed different types of microbiota samples (colorectal cancer tissue, non-cancerous tissues and feces) and overall have small sample sizes.

More studies aiming to analyze the association between human intestinal microbiota and surgical outcomes in larger samples, including consecutive patients undergoing colorectal surgery, are warranted. Therefore, we conducted this prospective study that analyzed the composition of two intestinal microbiota: mucosa associated microbiota (MAM) and luminal microbiota (LM) in consecutive colorectal cancer patients undergoing surgery. The primary endpoint of this study was to evaluate the potential impact of dysbiosis on postoperative course by comparing microbiota diversity and composition between patients with and without postoperative surgical complications. We also aimed to support the previously described data that MAM and LM are two distinct and complementary ecosystems [13], therefore highlighting the importance of analyzing both microbiome types in clinical studies. Lastly, we compared microbiota between patients with and without known risk factor for the development of postoperative complications who had received neoadjuvant therapy.

## 2. Materials and Methods

### 2.1. Patients

We performed a prospective, monocentric pilot study including consecutive patients undergoing elective surgery for cancer located in the left colon, sigmoid colon or rectum between June 2019 up to December 2021. In this article we present the preliminary results from the first 25 consecutive patients included in the study from June 2019 up to May 2020.

The inclusion criteria were: (1) cancer diagnosis confirmed by biopsy from coloscopy or recto-sigmoidoscopy prior to the surgery; (2) surgical resection with colorectal or coloanal anastomosis with or without protective ileostomy; (3) age of 18 years or more; (4) intention to have curative surgery; in the case of synchronous metastasis, a possible curative surgical or radiological treatment for metastases had to be validated and planned. The exclusion criteria were: (1) emergency surgery; (2) antibiotic administration 30 days prior to the surgery; (3) any other surgical intervention 30 days prior to the colorectal surgery; (4) pregnancy or breastfeeding; (5) no consent to participate expressed by the patient. Patients who used drugs with a strong impact on intestinal microbiota, such as laxatives and antibiotics, on a regular basis were not included in the study.

After enrollment, patients received bowel preparation with polyethylene glycol (PEG) or sodium sulfate solution 2 days and peroral antibiotics 1 day prior to the surgery (500 mg of metronidazole at 14 h, 15 h and 22 h). Additionally, all patients received an enteral nutritional supplementation (Oral Impact^®^) 7–10 days before the surgery.

Patients’ demographic and clinical data, neoadjuvant therapy and operative variables were recorded. All patients were followed up to 90 days postoperatively. Postoperative complications including surgical site infection (SSI), anastomotic leakage (AL) and postoperative ileus (PI) were recorded and classified according to Clavien–Dindo classification [14]. Together, SSI and PI represent more than 50% of readmission cases [3], whereas AL is associated with increased mortality and poorer oncological outcomes after colorectal cancer (CRC) surgery [2,15]. SSI was defined as an infection that developed 90 days after surgery and was classified as either superficial incisional SSI, deep incisional SSI or organ/space SSI [16]. PI was defined as a presence of nausea and vomiting, an inability to tolerate oral dietary intake, abdominal distension and a delayed passage of flatus and stool or a need to introduce a nasogastric tube due to the symptoms described above. AL was defined as a defect in the bowel wall at the anastomotic site that lead to communication between the intraluminal and extraluminal compartments; a pelvic abscess in close proximity to the anastomosis was also considered to be AL [17]. Antibiotics were not routinely administered in the postoperative course. Only in cases of development of postoperative complications, was the introduction of antibiotic treatment considered and administered according to local hospital protocols and underlying pathology. Duration of hospital stay, events of readmission and histological reports were recorded at the end of the follow-up period.

The study was approved by the institutional clinical research unit of Henri Mondor Hospital (Project No. APHP190088) and the French National Person Protection Committee (No. IdRCB no 2019-A00055-52). The study was in accordance with the principles of the Declaration of Helsinki by the World Medical Association. The study is registered in the Clinicaltrials.gov registry (ID: NCT04005118).

### 2.2. Samples Collection

Two types of samples were collected during the study course: (1) fresh feces for LM were retrieved by the patient before mechanical bowel preparation through an OMNIgene.GUT fecal sample collecting kit, which was received at the lab 72 h after collection, and stored at −80 °C until DNA extraction; (2) colon mucosa for MAM analysis was collected from mechanical stapler resection lines in a case of colorectal mechanic anastomosis or proximal resection line of the surgical specimen in a case of coloanal manual anastomosis, which was received at the lab immediately after collection, and stored at −80 °C until DNA extraction.

### 2.3. DNA Extraction

DNA was manually extracted from mucosal and fecal samples using QIAamp PowerFecal Pro DNA kits (Qiagen S.A.S., Courtaboeuf, France) according to a standard protocol provided. Extraction steps were as follows: (1) cell lysis and homogenization using up to 250 μg of each sample; (2) DNA inhibitor removal; (3) binding of the DNA; (4) DNA purification and washing; (5) DNA elution. For details see Bergsten et al. [18]. The isolated DNA was stored at −20 °C until use.

### 2.4. DNA Sequencing and Data Analysis

After prokaryote DNA were checked for quantity and quality, V3–V4 16sRNA islet was amplified and submitted to the metagenomic pair-end sequencing procedure using the Illumina MiSeq next-generation sequencing (NGS) system. Two primer sequences were used in this polymerase chain reaction (PCR) step: 16S Amplicon PCR Forward Primer = 5′ (TCGTCGGCAGCGTCAGATGTGTATAAGAGACAGCCTACGGGAGGCAGCAG) and 16S Amplicon PCR Reverse Primer = 5′ (GTCTCGTGGGCTCGGAGATGTGTATAAGAGACAGGACTACCAGGGTATCTAATCC). The size and quality of each final sample was detected using automated electrophoresis (TapeStation, Agilent). Each sample was then quantified using an absorbance quantification method with UV–visible spectroscopy (Varioscan, ThermoFisher Scientific, Waltham, MA, USA). Final sequencing was performed through the pair-end sequencing by synthesis (SBS) method.

Quality analysis of the reads and pair-end read merging were performed using FastQC v0.11.9 (https://www.bioinformatics.babraham.ac.uk/projects/fastqc/ accessed on 29 November 2021) and FLASH2 v2.2.00 software [19], respectively. Quality trimming of the sequences and taxonomic assignment were performed using the Trimmomatic v0.38 program [20] and the DADA2 pipeline v1.6 [21], respectively. Final bacteria taxonomic identification was performed using the SILVA138 rRNA database [22].

An amplicon sequence variant (ASV) table was used for statistical analysis with the Shaman website from Pasteur Institute [23], which implements an analysis using generalized linear models (package DESeq2/R) with defined co-factors [24]. Beta diversity was measured using Bray–Curtis metrics and was visualized by using the principal coordinates analysis (PCoA) technique. A PERMANOVA test was performed to analyze the variances by using distance matrices with *p* values set to <0.05.

### 2.5. Statistical Analysis

Descriptive statistics were performed to describe the study population and to compare the groups with and without surgical complications. For bivariate two-sided comparisons, the chi-squared test or Fisher’s exact test were used for categorical variables, while the Mann–Whitney U test was applied for continuous variables. A *p* value < 0.05 was considered to be statistically significant. Statistical analyses were performed using IBM SPSS software.

## 3. Results

Over the reported study period, 25 patients were enrolled. The demographic, clinical and operative characteristics of the patients are summarized in Table 1. Overall, 8 patients (32%) developed postoperative complications; their data were compared to those who did not experience postoperative complications (*n* = 17). Among patients with postoperative complications, 6 out of 8 were classified as ASA III or IV that was significantly higher than those without complications (75% vs. 17.6%, *p* = 0.01). No other demographic, clinical or operative differences were found between groups. The use of drugs with a potential impact on microbiota (proton pump inhibitors and metformin) did not differ significantly between groups; none of the patients received probiotics in the perioperative period.

Neoadjuvant therapy was administered according to the French national guidelines [25]. A majority of the patients (56%) received neoadjuvant therapy: chemoradiotherapy (*n* = 11, 10 patients peroral capecitabine, 1 patient intravenous FOLFOX), radiotherapy alone (*n* = 2) and chemotherapy alone (*n* = 1, FOLFOX).

Postoperative variables are summarized in Table 2. Briefly, three patients developed more than one postoperative complication: in one case, postoperative ileus and urinary infection; in the second case, anastomotic leakage and postoperative ileus combined with perineal edema and urinary infection; in the third case, superficial surgical site infection and acute urinary retention. Patients with complications had a significantly longer hospital stay (15.75 vs. 9.26 days, *p* = 0.01) than patients with no complications. The other postoperative variables considered did not significantly differ between the two groups.

Colon mucosa samples were available from all 25 patients and fresh feces samples from only 20 patients (samples not provided by patients).

### 3.1. Prokaryote DNA Characteristics

After amplicon and index PCR, the mean DNA concentration of the 16S V3–V4 amplicons was 285.6 ng/μL (range 105.1–372.2 ng/μL). Randomly chosen samples were used for size and quality control of the V3–V4 islets amplified 16S fragments using gel electrophoresis. All fragments demonstrated migration and no fragmentation was observed; the size of the fragments varied around 550 bp. The average cluster density was 792 K/mm^2^. A Phred quality score of Q ≥ 30 was attributed to 89.9% of bases.

### 3.2. Metagenomic Analysis

#### 3.2.1. General Analysis

Analyses were performed using 45 files (25 samples from colon mucosa and 20 samples from fresh feces; the 5 remaining cases were not analyzed due to missing samples or failure of quality). The mean number of reads per file was 255,699.1 (SD ± 81,625.72). FastQC analysis showed a Q > 30 score to 95% of reads. FLASH2 analysis demonstrated end-pair merging for >90% of reads. In total, 12,085 amplicon sequence variants (ASVs) were identified. After identification with the SILVA138 rRNA database, these ASVs were assigned to 246 different genera.

#### 3.2.2. MAM versus LM Signatures

Comparing MAM and LM from all patients, PCoA demonstrated two distinct bacterial communities with statistically different compositions (*p* < 0.001) (see Figure 1).

Analysis was performed according to principal coordinate analysis (PCoA) that showed significant separation between the mucosa (MAM designed as colon) and fresh feces (LM designed as feces) (*p* < 0.001).

The abundance of 28 genera differed significantly between the MAM and LM. The 16 most abundant genera displaying a base mean ≥ 100 are illustrated in Table 3.

A higher alpha diversity was observed in LM compared to MAM.

Most abundant bacteria were significantly different between MAM and LM. *Enterococcus* and *Escherichia/Shigella* share many similar characteristics and are described as one taxonomic unit at the genus level. Log2 fold change value is calculated when comparing the base mean value in mucosal vs. fecal samples. MAM: mucosa associated microbiota, LM: luminal microbiota.

#### 3.2.3. Microbiota Analyses According to the Presence of Postoperative Complications

Beta diversity was lower in patients with postoperative complications than in patients without complications, for both MAM and LM.

When analyzing the MAM and LM samples together, no significant difference was found in bacterial composition between the two groups with and without postoperative complications. For further analysis of bacterial abundance, the mucosal and fecal samples were analyzed separately. Among the 25 MAM and 20 LM samples, 8 and 7 were from the complication group, respectively. Three variables—gender, age and body mass index (BMI)—were included in the analysis as confounding factors. The MAM and LM showed compositional differences when the complication and no-complication groups were compared. The distributions of the 12 most abundant genera in each sample type (MAM vs. LM) and group (postoperative complications vs. no complications) are illustrated in Figure 2a. The distributions of the differential abundances, including sample types and groups, are shown in Figure 2b. However, statistical analysis did not show a significant differential abundance between the complication and non-complication groups in either the MAM (*p* = 0.983) or LM (*p* = 0.472) (Figure 3).

For this article, a sub-group analysis result comparing each major surgical complication (AL, PI, SSI) to non-complication group is not reported due to the very small complication event number (AL = 1, PI = 4, SSI = 2). However, in a case of sufficient sample number, it will be reported after completion of the study with the intended 50 patients included.

#### 3.2.4. Effect of Neoadjuvant Therapy

Taking together all patients and analyzing the MAM and LM samples together, those patients who had received neoadjuvant therapy (*n* = 14) showed a significant (*p* = 0.04) separation when compared to those without neoadjuvant therapy (*n* = 11), as assessed by PCA analysis (Figure 4). Bacteria abundance at the genus level was further analyzed separately in the mucosal (MAM) and fecal (LM) samples. Focusing on genera with mean abundance higher than 100 (mean base ≥100/sample) two bacteria genera, *Lactobacillus* and *Dialister*, were significantly less abundant in the neoadjuvant sub-group of patients (*p* = 0.047 and *p* = 0.048, respectively) in the mucosal samples (MAM), while in the fecal samples (LM), three genera showed significant difference in abundance between sub-groups: *Prevotella* and *Megasphaera* were less abundant (*p* = 0.0096 and *p* = 0.0008, respectively), and *Erysipelatoclostridium* was more abundant (*p* = 0.048) in the neoadjuvant sub-group.

## 4. Discussion

In our study, we performed metagenomic analysis to characterize the bacterial composition in the colonic mucosa (MAM) and feces (LM) in CRC patients. In our preliminary analysis of 25 consecutive patients we observed that the MAM and LM compositions differed significantly in our population. We showed that the neo-adjuvant chemoradiotherapy, which was administered to the majority (56%) of included patients, had a significant impact on microbiota composition. We further compared the MAM and LM in patients who developed 90-day postoperative complications (*n* = 8) to patients without postoperative complications (*n* = 17) and observed no significant difference in microbiota composition between these two groups.

Although impact of microbiota dysbiosis has been designed as the primary endpoint to explain complications in the postoperative course, we could not achieve it likely due to the lack of power. The small sample size should be accounted for that. Taxonomic analysis of the 25 MAM samples and 20 LM samples revealed that these are two distinct microbial communities with the presence of 28 genera differing significantly between both sample types. Ringel et al. characterized mucosal and fecal samples in 24 healthy individuals [13]. In their study, 10 genera were found in the MAM and were not present in the LM. The present study has endorsed similar observations in the CRC patient population with distinct genera found in MAM and LM. This suggests that by analyzing only LM, various potential mucosa-adherent bacteria with potential impact on the mucosa behavior may be missed.

Neoadjuvant therapy is a standard approach for locally advanced rectal cancer and metastatic CRC and has been increasingly used in locally advanced colon cancer [25,26]. In the present study, most of the included patients (*n* = 14, 56%) received neoadjuvant therapy. Analysis revealed that the patients’ intestinal microbiota composition (MAM and LM analyzed together) differed significantly between the two groups, suggesting that neoadjuvant therapy may exert an impact on microbiota compositional changes. This has been demonstrated in the case of pancreatic cancer; Goel et al. reported alterations of biliary microbiota in patients receiving neoadjuvant therapy before pancreatoduodenectomy [27]. Montassier et al. showed significant changes in LM composition that were associated with gastrointestinal mucositis and lower microbiota diversity in patients receiving chemotherapy for non-Hodgkin’s lymphoma [28]. To the best of our knowledge, no similar studies have been published regarding CRC. Based on our observations, which revealed that patients with and without neoadjuvant therapy showed significantly different microbiota signature, we suggest that the chemoradiotherapy constitutes a confounding factor in metagenomic analysis in CRC patients.

Although the patients with postoperative complications showed a trend of difference to bacterial abundances compared to those without complications, neither MAM nor LM compositions differed significantly between these groups. This suggests bias due to the heterogeneity of dysbiosis related to each complication; alternatively, due to the small sample size of the present series.

Currently, very few published human studies have analyzed the intestinal microbiota associated with postoperative complications after CRC surgery; most of them focus on anastomotic leakage (LA) [11,12,29] and one on postoperative ileus [11]. Praagh et al. [9] compared 11 patients with AL to 49 patients without AL. They suggested that a higher abundance of *Bacteroidaceae* (*Bacteroidetes phylum*) and *Lachnospiraceae* (*Firmicutes phylum*) families in the MAM in association with a decreased microbiota diversity could predict an increased risk of the development of AL. The authors hypothesized that this observation could be due to the mucin-degrading properties of the detected bacteria in the study samples. Palmisano et al. [10] observed dysbiosis in the LM samples from 5 patients with AL after CRC surgery and speculated that the development of AL could be related to a compositional disbalance between potentially pathogenic genera such as *Acinetobacter* and *Hafnia* (*Proteobacteria phylum*) and “protective” ones such as *Barnesiella* (*Bacteroidetes phylum*). Mima et al. [30] analyzed CRC and non-cancerous colon tissues in 256 patients and focused exclusively on bacteria that had previously been shown to be associated with CRC in animal models. They used quantitative PCR to measure relative amounts of *Fusobacterium nucleatum*, pks-positive *Escherichia coli*, *Enterococcus faecalis*, and *Bifidobacterium* and revealed that *Bifidobacterium* genus high cases were associated with an increased risk of anastomotic leakage (29 patients). Finally, Jin et al. [11] reported the results on cancerous MAM with potential association with perioperative ileus. They reported that patients with lower abundance of *Faecalibacterium* had an increased risk of early postoperative ileus and that *Faecalibacterium* could potentially be used as a biomarker for predicting this complication. However, the sample size for the validation of this hypothesis included only 6 patients with early postoperative ileus.

Despite promising information from existing human studies on microbiota changes and their potential association with postoperative complications, we feel that at this stage we cannot produce a conclusion because of several limitations: (1) studies included populations with high diversity by including benign and malignant pathologies; (2) discrepancy of microbiota types (MAM and LM) and tissues (cancerous and non-cancerous); (3) studies including the present one included small sample sizes, particularly those with a complicated postoperative course; (4) limitations related to fecal and tissue DNA sequencing techniques and assignation. In our study, the taxonomic analysis was performed at the genera level, that is, the suggested level when sequencing the 16S V3–V4 region. V3–V4 does not allow for some species to be distinguished within the same genus and can potentially merge them together, leading to a misinterpretation of the results. However, we acknowledge that distinguishing different species and bacterial strains can be clinically relevant when studying the pathogenesis of a particular condition [18].

We also recognize that the study design contains two factors that have a significant impact on colonic microbiota—preoperative mechanical bowel preparation and peroral antibiotics. It has been widely reported that these two interventions considerably change microbiota composition and diversity [8,31]. However, these interventions are widely applied before colorectal surgery in order to carry out a “clean” intervention that minimizes the risk of fecal contamination of the operative field, particularly during the anastomosis preparation. Most recent publications have reported that preoperative bowel preparation decreases a risk of development of postoperative complications after colorectal resections [29,32]. Despite the known impact on colonic microbiota, our study, therefore, was designed to provide the best perioperative care to included patients. Moreover, all the included subjects received preoperative mechanical bowel preparation and peroral antibiotics to create a homogenous population regarding these possible confounders.

The main limitation of our report is an insufficient statistical power due to a small sample size. However, these are preliminary results of the first 25 patients and complete analysis will be performed with the full patient population included at the end of the study.

## Figures and Tables

**Figure 1 microorganisms-10-00041-f001:**
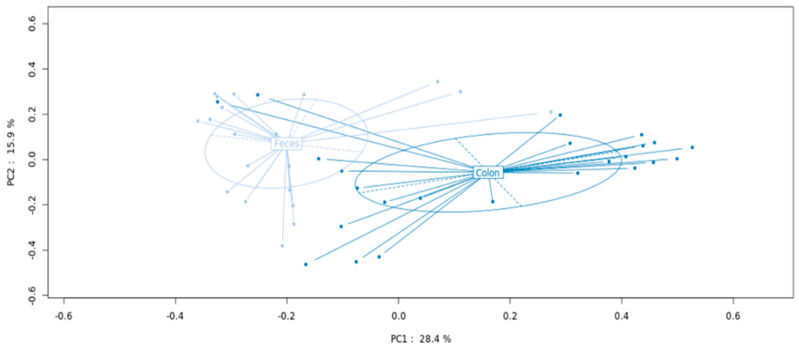
MAM and LM metagenomic profiles in patients with colorectal cancer undergoing curative surgery.

**Figure 2 microorganisms-10-00041-f002:**
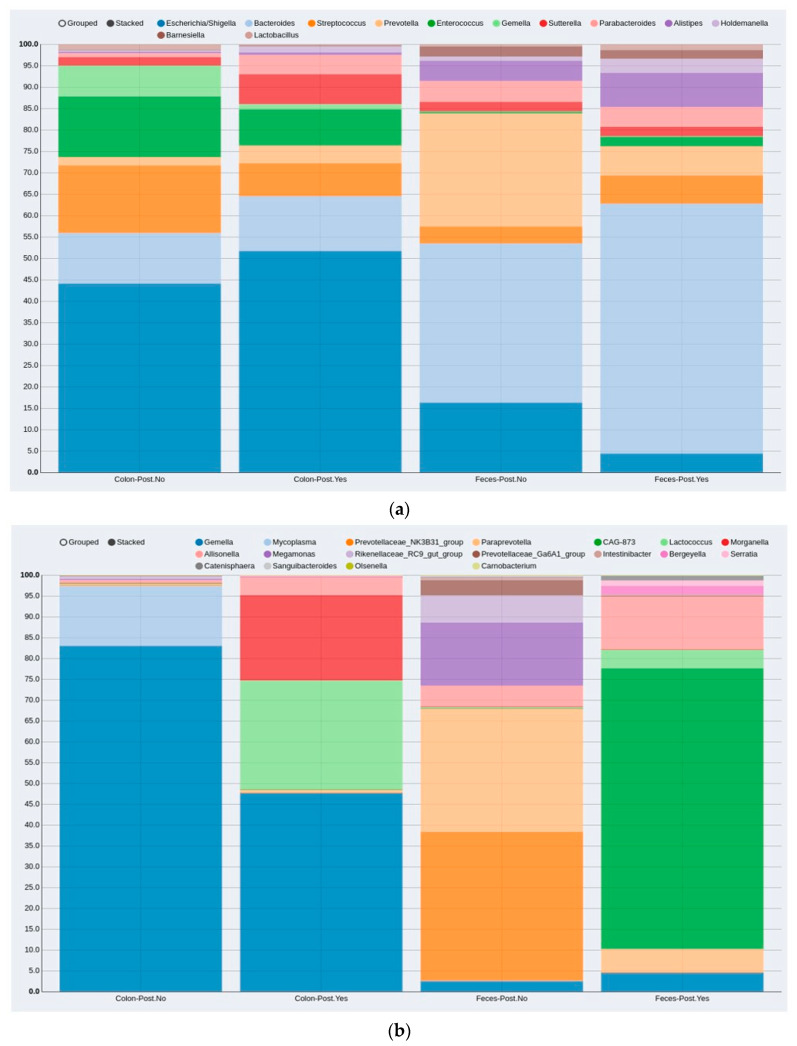
Compositional distribution of bacteria genera in patients without and with post-surgery complications according to MAM (colon) and LM (feces) sample types: (**a**) compositional distribution of the 12 most abundant genera; (**b**) differential abundances (genera that differ in abundance between the MAM and LM). MAM: colon-mucosa associated microbiota, LM: feces-luminal microbiota, Post.No: no postoperative complications, Post.Yes: postoperative complications.

**Figure 3 microorganisms-10-00041-f003:**
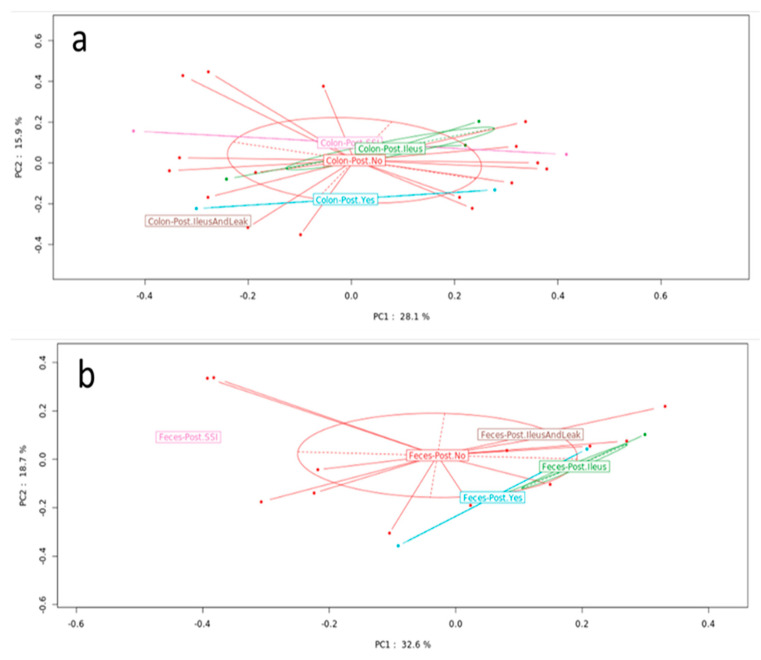
Microbiota composition in patients with and without postoperative complications. PCoA showed no significant difference between groups in the (**a**) MAM ((**a**), *p* = 0.983) or (**b**) LM ((**b**), *p* = 0.472) samples. PCoA: principal coordinate analysis, MAM: mucosa associated microbiota, LM: luminal microbiota.

**Figure 4 microorganisms-10-00041-f004:**
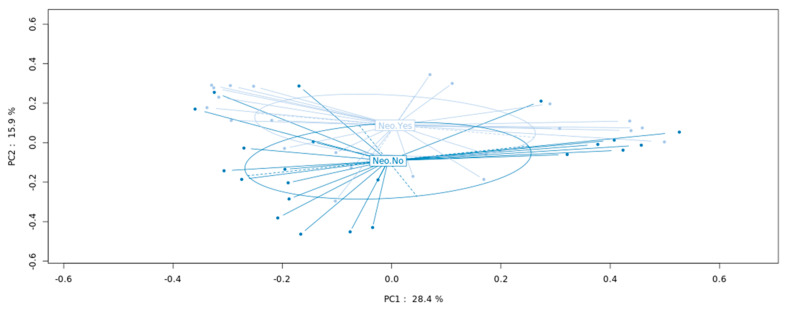
Microbiota composition in patients with and without neoadjuvant therapy. PCoA showing a significant difference between the bacterial composition in patients who had received the neoadjuvant therapy versus those with no neoadjuvant therapy (*p* = 0.04). PCoA: principal coordinate analysis.

**Table 1 microorganisms-10-00041-t001:** Demographic, clinical and operative variables in consecutive patients (*n* = 25) with colorectal cancer undergoing surgery.

	All Patients (*n* = 25)	Complications (*n* = 8)	No Complications (*n* = 17)	*p* Value
**Demographic and clinical variables**				
**Gender** (male/female, *n* (%))	14 (56)/11 (44)	7/1	7/10	0.042
**Age** (years, mean (SD))	68.64 (±11.64)	69.63 (±8.31)	68.18 (±12.89)	0.977
**BMI** ≥ 30 (kg/m^2^, *n* (%))	6 (24)	2	4	0.370
**Weight loss** ^$^ (*n* (%))	3 (12)	1	2	1
**ASA score** (*n* (%))				
I-II	16 (64)	2	14	0.01
III-IV	9 (36)	6	3	
**CCI scores** (*n*(%))				
3 4 5 ≥6	3 (12) 5 (20) 4 (16) 13 (52)	0 1 3 4	3 4 1 9	0.374
**Previous abdominal operations** (*n* (%))	11 (44)	2	9	0.234
**Drug use with potential impact on microbiota** * (*n* (%))	6 (24)	4	2	0.059
**Cancer location** (*n* (%))				0.941
Left colon	3 (12)	1	2	
Sigmoid colon	4 (16)	1	3	
Colorectal junction	2 (8)	1	1	
Rectum	16 (64)	5	11	
**Synchronous metastases** (*n* (%))	3 (12)	1	2	1
**Neoadjuvant therapy** (*n* (%)) Chemoradiotherapy Chemotherapy	14 (56) 11 (44) 1 (4)	4 3 0	10 8 1	1
Radiotherapy	2 (8)	1	1	
No neoadjuvant therapy	11 (44)	4	7	
**Prealbumin levels** (ng/L, mean (SD))	278 (±68.87)	277.75 (±41.61)	277.6 (±79.68)	0.975
** *Operative variables* **				
**Type of anastomosis** (*n* (%))				0.948
Colorectal	15 (60)	5	10	
Coloanal delayed ^#^	6 (24)	2	4	
Coloanal	4 (16)	1	3	
**Ileostomy** (*n* (%))	11 (44)	4	7	1
**Surgical approach** (*n* (%))				0.637
Robotic	17 (68)	5	12	
Laparoscopic	7 (28)	3	4	
Open	1 (2)	0	1	
**Conversion to open** (*n* (%))	0	0	0	NA
**Operative time** (min, mean (SD))	357.4 (±91.44)	383.75 (±96.23)	345 (±86.36)	0.344
**Transfusions** (*n* (%))	0	0	0	NA

The cohort is divided into patients with (*n* = 8) and without (*n* = 17) postoperative complications over a 90-day follow-up period. BMI: body mass index, ASA: American Society of Anesthesiology, CCI: Charlson Comorbidity Index, NA: not applicable. ^$^ >10% of body weight in last 6 months, * proton pump inhibitors, metformin, ^#^ coloanal anastomosis constructed 1 week after the index operation with resection.

**Table 2 microorganisms-10-00041-t002:** Postoperative variables over a 90-day post-surgery follow-up period.

	All Patients (*n* = 25)	Complications (*n* = 8)	No Complications (*n* = 17)	*p* Value
**Postoperative variables**				
**Overall complications** (patients, *n* (%))	8 (32)	8	0	NA
PI	4			
SSI	2			
AL	1			
Anastomosis bleeding	1			
Perineal edema	2			
Acute urinary retention Urinary infection	2			
	1			
**Clavien–Dindo ≥3**	2	2	0	NA
**Mortality**	0	0	0	NA
**Hospital stay** (days, mean (SD))	11.36 (±4.75)	15.75 (±4.15)	9.29 (±3.46)	0.01
**Reoperation**	0	0	0	NA
**Readmission** (*n* (%))	2 (8)	2	0	0.547

PI: postoperative ileus, SSI: surgical site infection, AL: anastomotic leakage.

**Table 3 microorganisms-10-00041-t003:** Differential abundant bacterial genera in consecutive patients (*n* = 25) with colorectal cancer undergoing surgery.

Genus	Base Mean	log2 Fold Change	Adjusted *p* Value
**Significantly more abundant in MAM**			
*Escherichia/Shigella*	42,410.3	2.534	0.005
*Streptococcus*	13,446.3	2.522	0.001
*Enterococcus*	11,135.19	4.805	0.000003
*Granulicatella*	783.79	4.59	8.567^−7^
*Actinomyces*	111.03	2.071	0.04989
**Significantly more abundant in LM**			
*Bacteroides*	25,946.21	−1.26	0.008
*Alistipes*	2395.64	−3.415	5.466^−14^
*Barnesiella*	959.38	−2.766	0.00069
*Odoribacter*	587.57	−1.611	0.041
*Phascolarctobacterium*	499.36	−2.987	0.001
*Bilophila*	344.62	−2.577	0.0008
*Butyricimonas*	341.26	−2.666	0.003
*Acidaminococcus*	277.84	−3.423	0.024
*Desulfovibrio*	272.76	−3.694	0.000013
*Paraprevotella*	241.54	−3.499	0.023
*Succiniclasticum*	166.3	−4.527	0.041

## Data Availability

The data presented in this study are available on request from the corresponding author. The data are not publicly available due to ethical and privacy restrictions.

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
