# Peer review of "Human Colonic Microbiota and Short-Term Postoperative Outcomes in Colorectal Cancer Patients: A Pilot Study"

_microorganisms, 2021, doi:10.3390/microorganisms10010041_

Round 1

Reviewer 1 Report

Manuscript entitled "Human colonic microbiota and short-term postoperative outcomes in colorectal cancer patients: preliminary results " submitted by Lauka et al. gained my attention. It is dealing with very interesting and current topic - the role of microbiota in health and dieases. This paper focus on microbiota composition in patients suffering from colorectal cancer. It is a pity that only data on microbiota composition from 25 patients are presented without any other patients characteristics (e.g. cytokines levels, tight junction proteins associated with barrier functions, antimicrobial peptides, etc.). I am suggesting to reformat this paper to short communication. In addition to that, patient cohort should be better described (did they used probiotics, diary products etc.). The mode of delivery impacts the microbiota composition as well. This should be included in population characteristics and discussed in discussion. 

Minor comments

Remove double space (e.g. lines 69; 74; 82 etc.)

and should not be in intalic (e.g. lines 235; 237)

Author Response

Review report, Reviewer 1

Manuscript entitled "Human colonic microbiota and short-term postoperative outcomes in colorectal cancer patients: preliminary results " submitted by Lauka et al. gained my attention. It is dealing with very interesting and current topic - the role of microbiota in health and diseases. This paper focus on microbiota composition in patients suffering from colorectal cancer.

It is a pity that only data on microbiota composition from 25 patients are presented without any other patients characteristics (e.g. cytokines levels, tight junction proteins associated with barrier functions, antimicrobial peptides, etc.).

Answer:

Indeed, this pilot study was designed to focus on metagenomic analysis purely. We agree that there would be an interest in the future studies to combine metagenomic data with cytological, molecular and biological analysis to study particular pathways  of microbiota – colonocyte interaction and, eventually, its role in the development of different postoperative complications. However, we find that at this stage it is a priority to define specific microbiota compositional changes and to detect potential bacterial genera/ species involved in the pathogenesis of each complication separately (surgical site infection, postoperative ileus, anastomotic leakage) before studying its functional pathways.

I am suggesting to reformat this paper to short communication.

Answer:

The authors would prefer to present the data in a form of an original article for several reasons: 1) the study design and the results from the population of 25 patients already at this point provide a valuable data regarding human intestinal microbiota and its association with postoperative complications – statistically significant decrease of diversity in a case of postoperative complications (potentially a preoperative/ operative marker of increased risk for the complications), significant impact of radiotherapy on microbiota composition (to our best knowledge, this is a first human study to present this data in a case of colorectal cancer), confirmation that MAM and LM are two distinct microbiota communities in a case of colorectal cancer (important for designing the future studies); 2) publishing an original article of the subject draws an attention to the lack of existing data regarding human colonic microbiota and postoperative course and emphasizes the necessity for future studies;  3) based on the existing literature (Praagh et al., Surg Endosc, 2016 30:2259–2265; Palmisano et al., Updates Surg, 2020 Dec;72(4):1013-1022)  human studies presenting the preliminary results even with a small sample sizes have been published as original articles, therefore stressing the importance of this subject and encouraging future studies.   

In addition to that, patient cohort should be better described (did they used probiotics, dairy products etc.). The mode of delivery impacts the microbiota composition as well. This should be included in population characteristics and discussed in discussion.

Answer:

An information regarding has been added:

  1. Exclusion of patients using laxatives and antibiotics in lines 74-76.
  2. Patients receiving enteral nutritional supplementation in lines 78-79.
  3. No difference in the use of drugs that impact the microbiota and no use of probiotics in lines 142-143 + Table 1.
  4. A paragraph regarding preoperative bowel preparation and preoperative antibiotics and its impact on microbiota has been added in the Discussion (lines 317 – 326).

No data regarding the dietary habits of included subjects was recorded.

Minor comments

Remove double space (e.g. lines 69; 74; 82 etc.)

Double spacing has not been used.

Should not be in italic (e.g. lines 235; 237)

Italic has been removed in lines 140, 141, 191, 192, 235, 237, 238, 282, 287, 290, 29, 293, 294.

Reviewer 2 Report

microorganisms-1510530-peer-review-v1

This is interesting study, evaluating the changes in microbial postoperative status, and try to find relations between different factors. In my opinion this manuscript can be suggested for publication, however, some adjustments and clarifications need to be taken in consideration by the authors.

Style of the papers needs to be adjusted to the recommendations from the journal. Reference list needs to be adjusted as well, taking care for the details.

Line 14. End of the sentence, change to "."

According to my knowledge, abstract do not need to be fragmented in Background, Methods, Results, Conclusions. Please check instructions for authors and adapt the abstracts to the recommendations from the journal.

Ln19: Maybe add information about if these patients were receiving different antibiotics treatments in postoperative recovery.

Please, all abbreviation needs to be explained.

Ln66-69: Please, check and correct. From one side you stated that this is study from June 2019 till December 2021, but from other side up to May 2020. Please, check and correct or explain better.

Ln76: (500 mg of metronidazole at 2 pm, 3 pm and 22 pm)." is this correct? For me, as not medical doctor is a bit confusing expressing time of administration of the antibiotic as present. Can you express yourself in better way?

Ln80: Please, correct to "....cases [17],"

Ln138: 75% vs. 17.6%,

Maybe authors will need to provide information if the patience were received any antibiotics, provide the list of applied antibiotics, if this was standard procedure, or were any differences?

Author Response

Review report, Reviewer 2

This is interesting study, evaluating the changes in microbial postoperative status, and try to find relations between different factors. In my opinion this manuscript can be suggested for publication, however, some adjustments and clarifications need to be taken in consideration by the authors.

Style of the papers needs to be adjusted to the recommendations from the journal. Reference list needs to be adjusted as well, taking care for the details.

Paper, including reference list,  has been adjusted following Microorganisms guidelines for authors.

Line 14. End of the sentence, change to "."

Changes have been made.

According to my knowledge, abstract do not need to be fragmented in Background, Methods, Results, Conclusions. Please check instructions for authors and adapt the abstracts to the recommendations from the journal.

Changes have been made and the headings have been removed.

Ln19: Maybe add information about if these patients were receiving different antibiotics treatments in postoperative recovery.

No routine administration of antibiotics was used. The complementary information is added in lines 93-95.

Please, all abbreviation needs to be explained.

Additional explanation of abbreviation has been added in the abstract, line 155, line 214. After careful revision, no other abbreviation without explanation have been noticed.

Ln66-69: Please, check and correct. From one side you stated that this is study from June 2019 till December 2021, but from other side up to May 2020. Please, check and correct or explain better.

It has been stated that the whole study period was from June 2019 up to December 2021 and that in this article we report only the results from first 25 consecutive patients that has been included from June 2019 up to May 2020.

It has been stated later on (lines 236-237, lines 316-318) that full analysis and additional information will be reported after inclusion of full patient population (all patients included in the study until December 2021).

Ln76: (500 mg of metronidazole at 2 pm, 3 pm and 22 pm)." is this correct? For me, as not medical doctor is a bit confusing expressing time of administration of the antibiotic as present. Can you express yourself in better way?

The timing has been changed to 14 h, 15 h and 22 h.

Ln80: Please, correct to "....cases [17],"

Changes have been made.

Ln138: 75% vs. 17.6%,

Changes have been made.

Maybe authors will need to provide information if the patience were received any antibiotics, provide the list of applied antibiotics, if this was standard procedure, or were any differences?

Information regarding no routine administration of antibiotics postoperatively has been added in lines 93-95. In cases of postoperative complications, antibiotics were used according to standard local protocols.

We feel that no complementary information about postoperative antibiotics should be added in this article as: 1) sampling of LM and MAM for metagenomic analysis were done before any introduction of antibiotics; 2) no sampling was done after introduction of antibiotics, 3) recording of the event – development of postoperative complications (to form a patients group of postoperative complications) -was always done before any introduction of antibiotics.

Round 2

Reviewer 1 Report

The quality of the manuscript has been improved after incorporation of reviewers´s comments and suggestions. Possibly, the role and recommendation of the probiotics administration can be discussed in the section Discussion and its possible role on microbiota composition in the patients. 

Minor comments 

Names of bacterial species and strains should be in intalic.

Author Response

Dear reviewer. The authors appreciate your suggestions and remarks. You have suggested:

Possibly, the role and recommendation of the probiotics administration can be discussed in the section Discussion and its possible role on microbiota composition in the patients. 

Answer: Authors acknowledge that the use of probiotics is a practice with potentially beneficial impact on postoperative period after elective colorectal surgery. Recent systemic reviews and meta-analysis have revealed that probiotics seem to decrease infectious postoperative complications (Darbandi et al., Clin Nutr. 2019; Liu et al., Gastroenterol Res Pract. 2017).

However, we don't feel that at this stage a discussion about the topic should be included in our article. None of included patients received probiotics before nor after the surgery (unlike mechanical bowel preparation and peroral antibiotics that have been discussed) making this subject inconsequent to presented population, methods and results in our study.

We agree that in cases where probiotics are included in the perioperative standard of care, their impact on microbiota should be discussed. In future studies probiotics could be proposed in the perioperative protocols.

Names of bacterial species and strains should be in italic.

Answer: All bacterial names on family level and below has been changed to italic.